# Effects of Subanesthetic Intravenous Ketamine Infusion on Stress Hormones and Synaptic Density in Rats with Mild Closed-Head Injury

**DOI:** 10.3390/biomedicines13040787

**Published:** 2025-03-24

**Authors:** Martin Boese, Rina Berman, Haley Spencer, Oana Rujan, Ellie Metz, Kennett Radford, Kwang Choi

**Affiliations:** 1Daniel K. Inouye Graduate School of Nursing, Uniformed Services University, Bethesda, MD 20814, USA; martin.boese@usuhs.edu (M.B.); kennett.d.radford.mil@health.mil (K.R.); 2Center for the Study of Traumatic Stress, Uniformed Services University, Bethesda, MD 20814, USA; rina.berman.ctr@usuhs.edu; 3Program in Neuroscience, Uniformed Services University, Bethesda, MD 20814, USA; haleyspencer121@gmail.com (H.S.); oana.rujan@usuhs.edu (O.R.); 4Department of Chemistry and Biochemistry, University of Maryland, College Park, MD 20742, USA; elliemetz04@gmail.com; 5Department of Psychiatry, F. E. Hébert School of Medicine, Uniformed Services University, Bethesda, MD 20814, USA

**Keywords:** ketamine, mild traumatic brain injury, stress hormone, synaptic density, rat, CHIMERA, prefrontal cortex, hippocampus

## Abstract

**Background:** Every year, over 40 million people sustain mild traumatic brain injury (mTBI) which affects the glucocorticoid stress pathway and synaptic plasticity. Ketamine, a multimodal dissociative anesthetic, modulates the stress pathway and synaptic plasticity. However, the effects of post-mTBI ketamine administration on plasma stress hormones and brain synaptic plasticity are largely unknown. **Methods:** Adult male Sprague-Dawley rats with indwelling jugular venous catheters sustained mTBI with the Closed-Head Impact Model of Engineered Rotational Acceleration (CHIMERA) in a single session (3 impacts × 1.5 J). One hour later, rats received intravenous (IV) ketamine (0, 10, or 20 mg/kg, 2 h). Catheter blood samples were collected for plasma corticosterone and progesterone assays. Brain tissue sections were double-labeled for presynaptic synapsin-1 and postsynaptic density protein 95 (PSD-95). Utilizing the Synaptic Evaluation and Quantification by Imaging Nanostructure (SEQUIN) workflow, super-resolution confocal images were generated, and synapsin-1, PSD-95, and synaptic density were quantified in the CA1 of the hippocampus and medial prefrontal cortex (mPFC). **Results:** IV ketamine infusion produced biphasic effects on corticosterone levels: a robust elevation during the infusion followed by a reduction after the infusion. CHIMERA injury elevated progesterone levels at post-injury day (PID)-1 and reduced synaptic density in the CA1 at PID-4, regardless of ketamine infusion. Ketamine infusion increased synaptic density in the mPFC at PID-4. **Conclusions:** Mild TBI and IV ketamine modulate the stress pathway and synaptic plasticity in the brain. Further research is warranted to investigate the functional outcomes of subanesthetic doses of ketamine on stress pathways and neuroplasticity following mTBI.

## 1. Introduction

Traumatic brain injury (TBI) is a leading cause of death and disability. After TBI, secondary injury processes occur that are mediated by neuroinflammation and excitotoxicity. One downstream consequence of TBI is a disruption in synapse function [1], which may particularly affect synaptic plasticity in regions such as the prefrontal cortex (PFC) and hippocampus [2]. Mild traumatic brain injury (mTBI) is the most common form of TBI, accounting for 70–90% of reported cases [3]. Though most individuals recover, up to 15% of people develop long-term disability after mTBI [4], including emotional disturbances, cognitive deficits, and physical symptoms [5]. In preclinical studies, rodents with mTBI may experience behavioral symptoms such as motor and balance dysfunction [6], cognitive impairment [7], and enhanced fear memory [8]. These symptoms are associated with synaptic alterations commonly observed after mTBI, including reductions in synaptic proteins and dendritic length, arborization, and spine density [7,8]. Thus, mitigating synaptic alterations following mTBI may be central to therapies aimed at reducing mTBI-induced disability.

Ketamine is a non-competitive N-methyl-D-aspartate (NMDA) glutamate receptor antagonist, commonly used as an anesthetic and analgesic drug. Ketamine is highly useful in trauma settings because it does not produce respiratory depression or cardiac instability [9,10,11]. The multimodal properties of ketamine, including potential immunomodulatory effects [12], may be beneficial for treatment of mTBI. Ketamine also has been shown to be neuroprotective in preclinical models of post-traumatic stress disorder (PTSD) [13], major depression [14,15], and TBI [16]. Moreover, ketamine has been demonstrated to reverse synaptic density changes in animal models of PTSD and depression, such as stress-induced alterations in dendritic branching, length, and spine density [13,17]. Ketamine may reverse the effects of TBI on synapses, as a daily ketamine injection (10 mg/kg, intraperitoneal [IP]) increased dendritic arborization, length, and spine density in the hippocampus after a moderate weight drop injury in rats [16]. However, little is known about the effects of IV ketamine infusions on synaptic density in the PFC and hippocampus after mTBI.

Another downstream consequence of TBI is disruption of the hypothalamic–pituitary–adrenal (HPA) axis, which releases hormones such as cortisol under stressful conditions [18,19]. TBI is generally considered to impair HPA axis activity [20], leading to consequences such as unchecked inflammation and psychiatric sequelae [21]. However, depending on TBI severity and timing of sampling after injury, cortisol may be reduced or elevated [22,23]. Cortisol and its rodent equivalent, corticosterone (CORT), peak and trough within two hours of an acute stressor [19], and changes in cortisol may be observed as early as 0–3 h after TBI. Similarly, preclinical studies have found elevated levels of CORT as soon as one hour post-mTBI in rats [24]. Based on these previous observations, we aim to examine the effects of mTBI on CORT at acute timepoints post-injury. Other hormones released during stress include progesterone (PROG), the biochemical precursor to cortisol [18], and a known neuroprotective agent [25]. TBI has been shown to modulate PROG levels in humans [26] and rodents [27,28]. However, endogenous PROG levels have yet to be examined, specifically after a mild TBI, perhaps because its role as a stress hormone is less well-known. To our knowledge, this is the first study to examine the effect of a preclinical model of mTBI on both basal CORT and PROG levels.

Ketamine has gained favor as a trauma drug because it provides potent analgesia and anesthesia without compromising hemodynamic stability or respiratory drive [10,11], and ketamine does not increase intracranial pressure (ICP) as was previously feared [29,30]. However, ketamine’s effects on HPA axis activity in TBI patients have yet to be examined, despite the importance of the HPA axis in the management of stress and recovering normal physiologic function after TBI [21]. Ketamine is known to modulate the HPA axis, increasing circulating and salivary cortisol levels in healthy human volunteers [31,32,33]. Similarly, studies from our lab have demonstrated that IV ketamine increases plasma levels of CORT and PROG in rats during a two-hour infusion, which return to baseline two hours later [34,35,36], which is mirrored in some human studies [33]. However, it is likely that ketamine will differentially modulate HPA axis activity in mTBI patients compared to healthy subjects. The current study is the first to examine the effects of IV ketamine on CORT and PROG at acute timepoints after mTBI.

Although many previous mTBI studies have demonstrated the importance of synaptic plasticity, conventional experimental methods used in those studies have limitations. Electron microscopy is a gold standard for synaptic evaluation, but obtaining images and manually counting synapses is labor-intensive and expensive. Dendritic spine analysis offers improved efficiency but uses dendritic spine morphology to evaluate neuroarchitectural changes. Synaptic proteins may be quantified using western blot, but this does not provide information on the spatial relationship between pre- and postsynaptic puncta. Synaptic Evaluation and Quantification by Imaging Nanostructure (SEQUIN) is a novel method to evaluate synaptic density by measuring colocalizations between pre- and postsynaptic markers [37,38] and has previously been used to find cortical synapse loss after CHIMERA injury [38]. SEQUIN produces a reproducible, quantifiable representation of synaptic density with rapid results compared to conventional methods such as dendritic spine analysis with Golgi staining or electron microscopy.

Despite the ubiquitous nature of mTBI injuries and the prevalence of ketamine as a trauma analgesic and anesthetic, the effects of ketamine on stress hormone levels and synaptic density following mTBI have not been investigated. We have established a paradigm of IV ketamine infusion in freely moving rats [36,39] and combined this technique with the Closed-Head Impact Model of Engineered Rotational Acceleration (CHIMERA) to investigate the effects of mTBI and ketamine on inflammatory cytokines and behavioral outcomes [6]. Unlike most preclinical TBI models, which utilize a craniotomy or head restraint, the CHIMERA simulates free movement in a clinical mTBI by delivering a closed-head impact to an unrestrained head. The specific aims of the current study were (1) to evaluate the effects of CHIMERA and ketamine infusion on plasma stress hormone levels and (2) to determine the effects of CHIMERA and ketamine on synaptic density in the mPFC and CA1 of the hippocampus of rats.

## 2. Methods

### 2.1. Animals

Male Sprague Dawley rats (9 weeks old) were purchased with jugular venous catheters surgically implanted at Envigo Laboratories (Dublin, VA, USA) as previously described [6]. Following three days of acclimation, animals were randomly assigned to one of six groups: Sham-0, Sham-10, Sham-20, CHIMERA-0, CHIMERA-10, and CHIMERA-20 (N = 9–10 per group). Blood and brain tissue samples used in this study were collected during a previous study that reported behavioral and neuroinflammatory effects of CHIMERA and ketamine administration [6]. None of the data collected and analyzed in this study were reported in the previous study. The animal protocol was approved by the Uniformed Services University Institutional Animal Care and Use Committee (IACUC) and followed all applicable federal regulations governing the protection of animals used in research.

### 2.2. CHIMERA

The CHIMERA injury procedure was performed as previously described [6]. Before the CHIMERA procedure, each animal was anesthetized using isoflurane (5% for induction and 3% for maintenance) mixed with 100% oxygen. Each animal was then placed in a dorsal position in the CHIMERA device with adhesive straps holding the body on the platform. The head was centered over crosshairs on an aluminum plate, aligning the impact piston approximately over bregma. A hole in the plate allowed a 200 g piston to impact the head (1.5 J, 5.5 m/s velocity). CHIMERA animals received three consecutive impacts (5–10 s apart) in a single session. Sham animals underwent the same procedure except for actual impacts to the head. After the injury, animals were returned to their home cages and allowed to drink acetaminophen water (1 mg/mL) in one of the two bottles installed in each cage (the other bottle contained water) for one day.

### 2.3. IV Ketamine Infusion

Before the infusion, racemic (±) ketamine hydrochloride (100 mg/mL) (Covetrus, Dublin, OH, USA) was diluted in 0.9% sterile saline to 2 mg/mL. One hour after the CHIMERA or sham injury, animals received an IV (R,S)-ketamine (0, 10, or 20 mg/kg) infusion over a 2 h period. The ketamine doses were selected based on our previous studies showing analgesic [36] and immunomodulatory [6] effects. Animals first received a bolus of ketamine (1 mg/kg, IV) or saline before the infusion procedure (Med Associates Inc., St. Albans, VT, USA) using infusion pumps (Harvard Pump 11 Elite, Holliston, MA, USA) that used a 5 mL plastic syringe with a flow rate of 2.5 mL/kg/h. Syringes were connected to a fluid swivel (Instech, Plymouth Meeting, MA, USA) by polyurethane tubing. This tubing was encased in a metal spring-wire tether that was magnetically attached to the metal cannula on the exit port of the catheter between the rat scapulae. The tether system allowed free movement of the animals in the chambers during the infusion period. Each chamber was equipped with two infrared photobeams for real-time locomotor activity monitoring. After the infusion, animals were returned to their home cages. Catheter blood samples were collected at 1 h post-injury (immediately before the ketamine infusion), 3 h post-injury (immediately after the ketamine infusion), 5 h post-injury (2 h after the ketamine infusion), and 24 h post-injury.

### 2.4. CORT and PROG ELISA

Plasma CORT and PROG levels were measured using enzyme-linked immunosorbent assay (ELISA) kits (Arbor Assays, Ann Arbor, MI, USA) as previously described [35]. Blood samples were centrifuged at 4000 rpm for 10 min at 4 °C, and plasma was collected and stored at −70 °C. For the ELISA, a serial dilution of standard samples was prepared and added to a 96-well plate. According to the manufacturer’s protocol, a diluted plasma sample, antibody, and CORT or PROG conjugate were added into each well. The plate was covered with a plastic film and incubated on an orbital shaker at room temperature. After incubation, the plate was washed with wash buffer three times. TMB substrate was added to each well, and the plate was incubated for 30 min at room temperature before the stop solution was added. The optical density was read at 450 nm using an Infinite 200 Pro Microplate Reader (Tecan US, Morrisville, NC, USA).

### 2.5. Immunohistochemistry

Four days after the injury, rats were deeply anesthetized with isoflurane, verified by paw pinch. A trans-cardiac perfusion with 10% neutral-buffered formalin in phosphate-buffered saline (PBS) was performed using a peristaltic perfusion pump. The brain tissue was removed from the calvarium, post-fixed in 10% neutral-buffered formalin for 24 h, and cryoprotected with a 20% sucrose solution in PBS for three days. Brains were sectioned with a sliding frozen microtome (Lecia Biosystems, Nussloch, Germany), and sections (40 μm) were stored in cryoprotectant solution at −20 °C. The CA1 and mPFC were determined based on the bregma coordinates from the rat brain atlas [40]. Four sections of brain tissue containing CA1 and four sections containing the mPFC from each animal were used for brain tissue analyses to capture representative data from the CA1 and mPFC, with the average data of the four sections being used in the analysis for each brain region. Representative sections of mPFC and CA1 are shown in Figure 1A. Immunofluorescent double-labeling was carried out as previously described [37]. Brain tissue sections were washed with PBS on a shaker. After the final wash, samples were placed in a six-well plate filled with a blocking buffer of 20% normal goat serum (Vector Laboratories, Newark, CA, USA) diluted in PBS for one hour. After blocking, the primary antibody solution was added, consisting of rabbit anti-PSD-95 (Invitrogen, Waltham, MA, USA) and guinea pig anti-synapsin-1 (Synaptic Systems, Göttingen, Germany) in 10% normal goat serum plus 0.3% Triton X-100 (Dow Chemical, Midland, MI, USA), and was incubated for one day at 4 °C. After the primary antibody incubation, the sections were washed, then incubated in secondary antibody solution in 10% normal goat serum plus 0.3% Triton X-100 for four hours at room temperature. Brain sections were washed, mounted on clean microscope slides, and dried in a flat, dark location for approximately 10 min. While the samples were drying, mounting media were prepared by mixing AF300 and MWL488 (Electron Microscopy Sciences, Hatfield, PA, USA) in a 1:9 ratio, followed by vortexing and desktop centrifuging to remove air bubbles (5 min, 4000 rpm). Once the sections were dry, 75 μL of mounting media was placed on the sample, and high-precision 1.5 H coverslip glass (Marienfield, Lauda-Königshofen, Germany) was used to protect the samples. The prepared slides were stored and cured in a dark room for 3–7 days before imaging with a confocal microscope.

### 2.6. SEQUIN

Sections were imaged on a Zeiss LSM 980 confocal microscope equipped with Airyscan 2 (Zeiss Group, Oberkochen, Germany). Scan parameters were established using previously described SEQUIN techniques [37]. CA1 and mPFC regions of interest (ROIs) were targeted under 10× power using an epifluorescent light source and microscope oculars. Once the ROI was targeted, Zeiss 518 immersion oil was applied, and the area was re-targeted under 63× power. Once the final ROI was confirmed, confocal images were obtained via previously described experimental parameters [37]. Confocal microscope images were analyzed using Imaris software (ver. 10.0.1, Abingdon, UK). Images were converted to Imaris format and used for spot analysis of PSD-95 and synapsin-1 puncta. Source channels were set to 488 nm and 594 nm for synapsin-1 and PSD-95, respectively. Estimated XY diameter was set at 0.3 μm based on manual measurement of typical puncta size. Point spread elongation was set at 0.65 μm to account for the axial distortion of confocal rendered images. Background subtraction and quality filters were utilized to capture maximum puncta detection while eliminating signal noise. Parameters were saved for batch analysis across all samples. Following spot analysis, a presynaptic to postsynaptic puncta examination was conducted using the Imaris software. Imaris identified and quantified postsynaptic PSD-95 puncta center points with a Euclidean distance < 0.55 μm from presynaptic synapsin-1 puncta center points. Identified puncta pairs were divided by the scanned image volume to determine relative synaptic density within each volume of sample analyzed. Densities from the four brain sections for each region (one CA1 or mPFC per section) were averaged.

### 2.7. Statistical Analyses

Tests for normal distribution of data were performed using Shapiro–Wilk tests, and all data passed normality tests except PROG data at the 1 h post-injury time point. Thus, that dataset was analyzed using non-parametric statistics (Mann–Whitney U test), and all other datasets were analyzed with parametric statistics. Student’s *t*-test was used for 2-group comparisons, and a two-way ANOVA was used for main effects and interaction between CHIMERA injury and ketamine doses. Post hoc tests (Holm–Sidak multiple comparisons tests) were used following significant ANOVA effects. For CORT and PROG time course data, a mixed model was used with between-subjects factors (CHIMERA and ketamine) and a within-subjects factor (time). All data analyses and plotting graphs were carried out with GraphPad Prism (ver. 10.4.1). Significance was determined at *p* < 0.05.

## 3. Results

Figure 1A represents the overall study design and specific timelines of the experiments. Animals were subjected to the CHIMERA injury and, one hour later, received an IV ketamine infusion. Catheter blood samples were collected at multiple timepoints, and brain tissue was collected at PID-4. Figure 1B shows spontaneous locomotor activity in the first hour of ketamine infusion (0, 10, or 20 mg/kg). A two-way ANOVA indicated a significant main effect of ketamine on locomotor activity (F_(2, 52)_ = 5.702, *p* = 0.0058). Post hoc tests revealed that 10 mg/kg reduced locomotor activity compared to the saline and 20 mg/kg groups. Figure 1C shows spontaneous locomotor activity in the second hour of ketamine infusion. A two-way ANOVA indicated a significant main effect of ketamine on locomotor activity (F_(2, 52)_ = 7.265, *p* = 0.0017). Post hoc tests revealed that 20 mg/kg increased locomotor activity compared to the saline and 10 mg/kg groups. The saline group showed less activity in the second hour as compared to the first hour due to habituation to the chambers over the two-hour infusion period.

Plasma CORT levels at 1 h post-injury were not statistically different between sham (mean 130 ng/mL) and CHIMERA (mean 156 ng/mL) groups (*p* = 0.094). This suggests that CHIMERA may produce mild effects on plasma CORT elevation (approx. 20%). However, the IV ketamine infusion produced robust increases in plasma CORT levels (approx. 3–4-fold) when measured immediately after the infusion (Figure 2A). A mixed model analysis indicates significant main effects of ketamine (F _(2, 47)_ = 34.38, *p* < 0.0001) and time (F _(1, 46)_ = 287.9, *p* < 0.0001) as well as an interaction between ketamine and time (F _(2, 46)_ = 64.12, *p* < 0.0001) on CORT levels. Both 10 mg/kg and 20 mg/kg groups increased CORT levels compared to the saline group based on post hoc tests. At 2 h post-infusion, CORT levels were significantly reduced in the 10 mg/kg and 20 mg/kg groups compared to the saline group. Plasma CORT levels returned to the baseline at PID-1 with no statistical differences between any of the groups at this time point (Figure 2B).

PROG levels were not significantly different between sham and CHIMERA groups at 1 h post-injury based on a Mann–Whitney test (U = 340, *p* > 0.05). The time course effects of ketamine on PROG levels at 0 h and 2 h post-infusion are shown in Figure 3A. A mixed model analysis with CHIMERA and ketamine as between-subjects factors and time as a within-subjects factor indicated a significant main effect of time (F _(1, 47)_ = 20.75, *p* < 0.0001) on PROG levels. Thus, IV ketamine infusion did not elevate PROG levels after the infusion. However, CHIMERA injury significantly elevated plasma PROG levels at PID-1 compared to the sham groups (Figure 3B). There was a significant main effect of CHIMERA on PROG levels (F _(1, 46)_ = 8.747, *p* = 0.0049) at this time point. This suggests that CHIMERA injury may produce a delayed increase in plasma PROG levels, which may serve as a potential biomarker of mTBI.

Figure 4A shows a representative image of synaptic density (yellow spots) based on synapsin-1 (green) and PSD-95 (red) double-labeling in the mPFC. Synaptic density using presynaptic (synapsin-1) and postsynaptic (PSD-95) markers was quantified with the SEQUIN method. There were no effects of CHIMERA or ketamine on synapsin-1 levels in the mPFC (Figure 4B). A two-way ANOVA revealed a significant main effect of ketamine on PSD-95 levels in the mPFC (F _(2, 52)_ = 11.83, *p* < 0.0001), as shown in Figure 4C. There was a significant interaction between CHIMERA and ketamine on PSD-95 density (F _(2, 52)_ = 3.362, *p* = 0.0423). Post hoc tests revealed that the 20 mg/kg group is significantly different from the saline and 10 mg/kg groups. A two-way ANOVA indicated a significant main effect of ketamine (F _(2, 52)_ = 3.968, *p* = 0.024) and a trend of significant interaction between CHIMERA and ketamine (F _(2, 52)_ = 3.009, *p* = 0.058) on the synaptic density in the mPFC (Figure 4D). Post hoc tests revealed a trend toward significance between the saline and 20 mg/kg groups (*p* = 0.06) and the 10 mg/kg and 20 mg/kg groups (*p* = 0.05).

Figure 5A shows a representative image of synaptic density (yellow spots) based on synapsin-1 (green) and PSD-95 (red) double-labeling in the CA1 of the hippocampus. A two-way ANOVA indicated a significant main effect of CHIMERA on synapsin-1 density (F _(1, 52)_ = 8.098, *p* = 0.0006), as shown in Figure 5B. Post hoc tests revealed that the Sham-0 group was significantly different from the CHIMERA-20 group (*p* < 0.05). A two-way ANOVA indicated a significant main effect of ketamine on PSD-95 density (F _(2, 52)_ = 5.491, *p* = 0.006), as shown in Figure 5C. Post hoc tests revealed a significant difference between the CHIMERA-10 and CHIMERA-20 groups (*p* < 0.05). A two-way ANOVA indicates a significant main effect of CHIMERA on synaptic density (F _(1, 52)_ = 4.747, *p* = 0.033), as shown in Figure 5D. This indicates that CHIMERA injury reduced synaptic density in the CA1 of the rat hippocampus at PID-4.

## 4. Discussion

In the current study, the effects of CHIMERA and subanesthetic doses of ketamine infusion on plasma stress hormone levels and synaptic density in the mPFC and CA1 were investigated using a rat model of mTBI. A CHIMERA injury produced a delayed increase in PROG levels at PID-1, and a ketamine infusion produced biphasic effects of CORT levels: elevation immediately following the infusion and reduction two hours after the infusion. The CORT levels returned to normal at PID-1. CHIMERA injury reduced synaptic density in the CA1, while the ketamine infusion increased synaptic density in the mPFC at PID-4. However, there was no interaction between CHIMERA injury and ketamine infusion on synaptic density. To our knowledge, this is the first study reporting the role of CHIMERA and IV ketamine infusion on stress hormones and synaptic plasticity in key brain regions that are vulnerable to mTBI.

After CHIMERA injury, CORT levels were not significantly altered but followed a pattern of slight increase at 1 h, decrease at 3 h, increase at 5 h, and return to baseline at 24 h post-injury. A similar time course was reported after a moderate controlled cortical impact (CCI), which utilized repeated blood sampling at 1.5, 6, 12, 18, and 24 h post-injury and found increased plasma CORT levels at 1.5 and 6 h, which subsequently fell to baseline and remained unaltered up to 24 h post-injury [41]. Unlike in the current study, those increases in CORT were significant, which is presumably due to a more severe and invasive injury model. However, plasma or serum CORT increases have been previously reported even after mTBI. Rats exhibited increased plasma CORT levels at 6 and 24 h after a mild midline fluid percussion injury (MFP) [42]. A repeated mild projectile concussive impact (PCI) injury increased serum CORT levels at 1 h post-injury, which returned to baseline levels at 24 h post-injury in rats [24]. Despite being framed as an mTBI, the PCI injury produced a staggering array of pathological and behavioral changes such as inflammation, neurodegeneration, gait alterations, and neurobehavioral deficits [24]. In contrast, our previous CHIMERA study only reported rotarod deficits and axonal damage in the optic tract [6]. This reinforces that the current injury is very mild, which would explain a lack of significant effect on CORT levels.

The IV ketamine infusion produced a biphasic effect on CORT levels: a robust increase consistent with previous studies [34,35,36], followed by a reduction below baseline at 2 h post-infusion. The observed dip in CORT below saline levels after ketamine administration in our study may be explained by negative feedback, whereby increased CORT binds to glucocorticoid receptors (GRs) in order to inhibit the HPA axis and reduce CORT secretion [43,44]. It is interesting to note that IV ketamine infusion caused CORT reduction at this time point in both sham and CHIMERA animals. This resulted in a relative suppression of CORT responses to CHIMERA injury when a normal physiological profile would increase CORT levels. CORT dysfunction, especially long-term CORT suppression, has been linked to the development of PTSD-like symptoms in preclinical studies and clinical studies [45,46,47]. It is unclear whether an acute reduction in CORT following mTBI could result in subsequent pathological changes. Studies have shown that basal CORT suppression, rather than acute CORT shifts, is implicated in neurobehavioral responses [45,48], including in mTBI patients [49]. Therefore, implications of biphasic CORT responses following IV ketamine infusion should be further investigated in the context of mTBI.

Unlike CORT, which had non-significant fluctuations in the first five hours post-injury, plasma PROG levels were stable following CHIMERA up to 5 h post-injury. However, CHIMERA injury significantly elevated plasma PROG levels at PID-1 in adult male rats. This finding may indicate a compensatory mechanism of PROG, which is considered to have a neuroprotective role after TBI, including inhibition of inflammation, attenuation of excitotoxicity, bolstering of myelin repair, and release of neurotrophic factors [25]. Another study reported that plasma PROG levels were elevated at 24 h post-injury with a weight drop model in male mice [27]. Thus, a delayed elevation of plasma PROG following CHIMERA injury may serve as a potential biomarker for mTBI and guide future therapeutic options for patients with mTBI.

Ketamine is known to exert synaptogenic effects through intricate signaling cascades. Its initial action is disinhibition of glutamate signaling at the NMDAR, preferentially inhibiting NMDARs on inhibitory GABAergic interneurons [50], leading to a paradoxical enhancement of glutamate release [51]. Glutamate activates postsynaptic α-amino-3-hydroxy-5-methyl-4-isoxazolepropionic acid receptors (AMPARs), which in turn activate tropomyosin receptor kinase B (TrkB) for brain-derived neurotrophic factor (BDNF) to stimulate downstream synaptic protein synthesis and synaptogenesis [14,52].

In the mPFC, ketamine produced robust effects on synaptic density, with a 20 mg/kg dose increasing PSD-95 and synaptic density. This effect of ketamine in the mPFC is in line with the literature demonstrating that ketamine increases spine density, synaptic markers, and synaptogenesis in the PFC and was able to reverse behavioral changes due to depression and anxiety [53,54,55,56]. As such, it would be worth examining depressive- and anxious-like behaviors in these rats to examine a connection between increased synaptic density in the mPFC and behavioral improvements. Behavioral and emotional changes after mTBI may be related to a loss of synaptic or functional connectivity in the PFC. For instance, mice sustaining mild lateral fluid percussion (LFP) injury experienced impaired fear extinction, which was concurrent with reduced complexity of dendritic arborization in the mPFC [8]. Conversely, restoration of dendritic spine length in the PFC by ketamine was concurrent with restored fear-extinction learning in rats [13]. It has been suggested that ketamine may improve fear extinction by activating synaptic signaling pathways in the mPFC, such as the mTOR pathway [57]. In humans, therapeutic effects of ketamine include upregulating functional connectivity in the PFC, which is analogous to increased synaptic connectivity in rodents [53]. Connectivity in the mPFC is inversely correlated to major depression, anxiety, fatigue, and post-concussive symptoms in patients with mTBI [58]. In depressed patients, therapeutic response to ketamine was associated with greater global brain connectivity in the PFC [59]. Taken together, the ability of subanesthetic doses of IV ketamine after mTBI to increase synaptic density in the mPFC found in the current study may have behavioral implications.

The limited effect of ketamine on synaptic density in the CA1 of the hippocampus was unexpected, as ketamine is known to produce synaptic effects in the hippocampus [17,52,60]. In the mPFC, ketamine significantly increased synaptic density along with the postsynaptic protein PSD-95 but did not affect the presynaptic protein synapsin-1. However, in the CA1, the CHIMERA injury produced a significant decrease in synapsin-1, which was unaffected by ketamine. Interestingly, ketamine increased PSD-95 in the CA1, although this effect was not strong enough to counteract the CHIMERA injury effect on overall synaptic density. The hippocampus has several factors that make it particularly vulnerable to traumatic injury. The hippocampus is situated directly below the corpus callosum, which transmits shearing forces to the hippocampus during brain injury [61]. The hippocampus is also flanked by the lateral ventricles, fluid-filled structures with large protrusions, which may contribute to the strain experienced in the hippocampus during the TBI [61,62]. Additionally, hippocampal neurons may intrinsically be more susceptible to mechanical damage than cortical neurons, experiencing greater calcium release and cell death when mechanically stretched [63]. Thus, the lack of effects of ketamine on synaptic density in the CA1 may be due to the specific nature of CHIMERA injury, along with the anatomy of the hippocampus being particularly vulnerable to this type of damage.

Ketamine produced dose-dependent effects in several metrics, namely locomotor activity during the infusion and synaptic density changes after CHIMERA injury. In the first hour of the ketamine infusion, the 10 mg/kg dose decreased locomotor activity relative to both saline and the 20 mg/kg ketamine dose due to its sedative properties at the lower 10 mg/kg dosage. However, in the second hour of the infusion, the 20 mg/kg dose increased locomotor activity compared to saline and the 10 mg/kg dose. Locomotor activity in the saline group was lower in the second hour due to habituation to the infusion chambers, and activity was higher in the 20 mg/kg group due to its dissociative effects, which likely prevented behavioral habituation. In the current study, the 20 mg/kg dose produced synaptic effects in the mPFC, while the 10 mg/kg dose did not. This is consistent with our previous studies indicating dose-dependent effects of IV ketamine infusion on plasticity-related proteins [64,65]. However, the inability of the 10 mg/kg dose to affect synaptic density in the current study is surprising, as this dose has consistently produced behavioral and physiological effects in the previous studies [12,35,39]. In other studies, administering a 10 mg/kg dosage via the IP injection exerted synaptic effects, increasing dendritic spine density and pre- and postsynaptic markers [15,54,55,60]. In the interest of preserving the therapeutic effects of IV ketamine while minimizing its side effects (e.g., dissociation and hallucination), further research is warranted to investigate specific dose-dependent ketamine responses on synaptic density and mTBI-related behaviors.

The current study is not without limitations. Only one time point after CHIMERA injury was investigated for synaptic density (PID-4). This time point is supported by the literature: a time course study of synaptic protein levels in the rat hippocampus after cortical contusion revealed that synapsin-1 and PSD-95 reached their lowest levels at 4 days post-injury [66]. Additionally, a two-photon imaging study in mice revealed that ketamine (10 mg/kg, IP) formed nascent spines that generally disappeared within four days in the medial frontal cortex of mice [67]. This four-day time point may be significant as newly formed dendritic spines that persist past four days can form functional synapses [68]. Further, persistent dendritic spines may be implicated in the sustained antidepressant effects of ketamine [15]. Therefore, analysis of the synaptic density at PID-4 may provide insights on the lasting effects of ketamine. However, synaptic growth is a dynamic process, and using only one time point does not allow for a complete picture of ketamine’s dynamic effects. Ketamine (10 mg/kg, IP) has time-dependent effects on spine density, with increases at 24 h but decreases at 7 days in the CA3 of the hippocampus in mice [60]. While ketamine induced the growth of nascent spines, it also retracted apical dendrite tuft branches the day after administration in the medial frontal cortex of mice [67]. Additionally, plasticity potential, behavioral effects, and the formation of new, persistent spines occur across different time courses after ketamine administration, with the most plasticity occurring at more acute (within 12 h) time points [69].

The main reasons for administering ketamine at the one-hour post-injury time point were (1) to adjust differences between humans and rodents. Rodents have a short lifespan (approximately 2 years), so one hour in rats can be a lot longer in humans. (2) Ketamine is often given within a few hours after injury in emergent and battlefield conditions due to its cardiovascular stability and maintenance of respiratory drive. (3) The effects of ketamine on HPA axis stimulation are fast and transient, so we aimed to overlap the effects of mild TBI and ketamine on HPA axis stimulation in rats. Similarly, we did not examine later time points of CORT and PROG levels after injury or ketamine infusion. We did not analyze these biomarkers beyond the first day because CORT typically peaks and troughs within two hours of a stressor [19], and we have previously observed these transient effects of ketamine on CORT levels as well, increasing and decreasing in the hours following infusion [34,35]. However, TBI is known to affect CORT and PROG levels several days after the injury [70,71,72], and follow-up studies would determine if mTBI produced persistent changes in HPA axis function. With the knowledge that PROG increased one day after injury, further testing of these later time points is warranted.

Another limitation of the current study is the use of only male rats, given the previous reports on sex-dependent effects of IV ketamine [12,73]. A study of male and female rats receiving ketamine (2.5 or 5 mg/kg, IP) after isolation stress found that ketamine reversed spine density loss in the mPFC and restored synaptic protein levels in male rats only [56]. Ketamine had more robust effects on spine density and synaptic marker expression in the hippocampus and PFC of male mice compared to female mice [60]. In particular, ketamine produced glutamatergic bursting in the mPFC of male mice only, which may be one explanation for greater synaptic protein upregulation [60]. Thus, future studies should examine the synaptic effects of ketamine in both sexes at multiple time points after mTBI to allow for greater clinical translation.

In conclusion, subanesthetic doses of an IV ketamine infusion produced dose-dependent effects on locomotor activity and increased postsynaptic PSD-95 proteins and synaptic density in the mPFC. Additionally, ketamine produced biphasic effects on plasma CORT levels after the infusion. The CHIMERA injury produced a delayed elevation of plasma PROG levels at PID-1, which can serve as a potential biomarker for mTBI. CHIMERA injury also produced a significant reduction in synaptic density in the CA1, which was not restored by ketamine infusion. As some of the therapeutic effects of ketamine are thought to be mediated by restoration of synaptic plasticity in the brain, increased synaptic density in the mPFC following IV ketamine infusion may have clinical significance. As the mPFC is one of the major target regions of ketamine, this finding has significant implications for its therapeutic potential following an mTBI. Overall, the current investigation demonstrates the utility of combining a clinically relevant mild closed-head injury in rats, IV ketamine infusion, and super-resolution confocal microscopy with an efficient synaptic data analysis workflow for mTBI research.

## Figures and Tables

**Figure 1 biomedicines-13-00787-f001:**
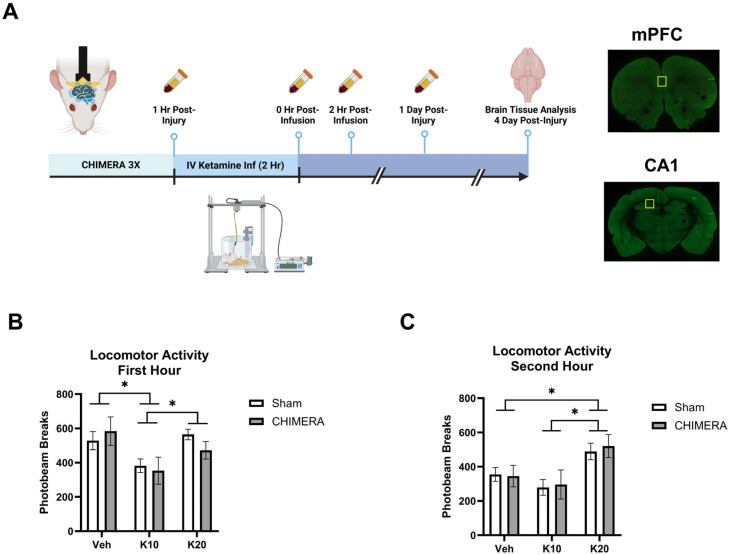
Experimental design and locomotor activity during ketamine infusion. (**A**) Experimental design indicating CHIMERA, IV ketamine infusion, catheter blood sampling, and brain tissue collection (Created in BioRender.com). Representative images of brain sections containing mPFC and CA1; the yellow square indicates the ROI. (**B**) Spontaneous locomotor activity during the first hour of IV ketamine infusion in rats. Ketamine 10 mg/kg (K10) infusion reduced activity compared to the saline vehicle (veh) and ketamine 20 mg/kg (K20) groups. * indicates a significant difference between the saline vehicle and K10 groups and the K10 and K20 groups (*p* < 0.05). (**C**) Spontaneous locomotor activity during the second hour of IV ketamine infusion in rats. Ketamine 20 mg/kg (K20) infusion increased activity compared to the saline and ketamine 10 mg/kg (K10) groups, as shown in * between the saline and K20 and the K10 and K20 groups (*p* < 0.05). N = 8–10 per group due to missing data (photobeam breaks counts malfunction in two animals).

**Figure 2 biomedicines-13-00787-f002:**
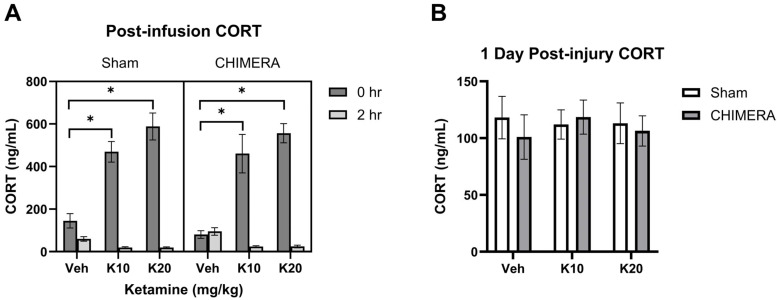
Plasma CORT levels following CHIMERA and IV ketamine infusion in rats. (**A**) CORT levels at 0 h and 2 h post-ketamine infusion (3 and 5 h post-CHIMERA). Ketamine produced robust effects on CORT elevation at 0 h followed by reduction at 2 h post-infusion (* *p* < 0.05). (**B**) CORT levels at 1 day after CHIMERA injury returned to normal, baseline levels in rats. N = 7–10 per group due to lack of blood samples and missing data.

**Figure 3 biomedicines-13-00787-f003:**
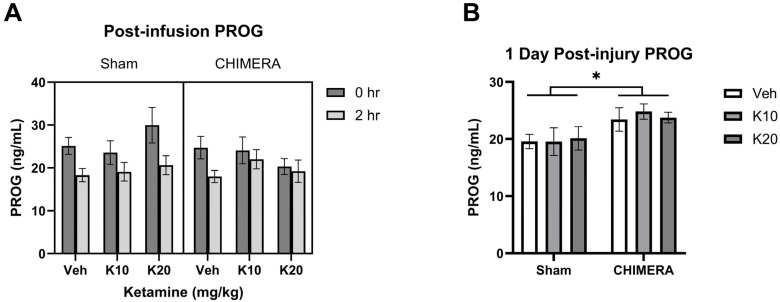
Plasma PROG levels following CHIMERA and IV ketamine infusion in rats. (**A**) PROG levels at 0 h and 2 h post-ketamine infusion (3 and 5 h post-CHIMERA). Ketamine did not alter PROG levels at these time points. (**B**) PROG levels at 1 day after CHIMERA injury. CHIMERA injury significantly elevated PROG levels at this time point (* *p* < 0.05). N = 7–10 per group due to lack of blood samples and missing data.

**Figure 4 biomedicines-13-00787-f004:**
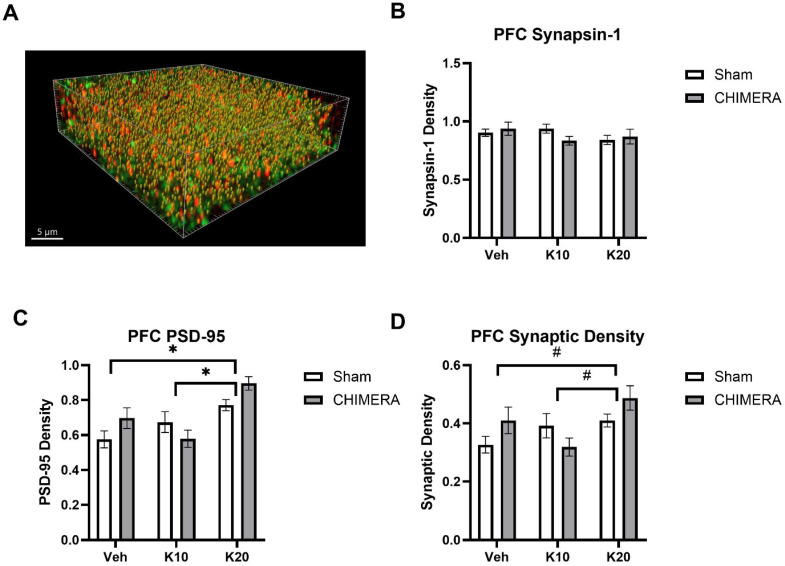
Synaptic density in the mPFC following CHIMERA injury and IV ketamine infusion in rats. (**A**) A representative Imaris image showing synapsin-1 (green), PSD-95 (red), and synaptic (gold) puncta in the mPFC. (**B**) CHIMERA and ketamine had no effects on synapsin-1 density in the mPFC (*p* > 0.05). (**C**) PSD-95 density in the mPFC. Ketamine 20 mg/kg increased PSD-95 density compared to the saline and ketamine 10 mg/kg groups (* *p* < 0.05). (**D**) Synaptic density in the mPFC. Two-way ANOVA indicated a significant main effect of ketamine on synaptic density in the mPFC. However, post hoc tests revealed trends of significance between saline and K20 (*p* = 0.06) and K10 and K20 (*p* = 0.05) groups, as indicated by # on the graph. N = 9–10 per group.

**Figure 5 biomedicines-13-00787-f005:**
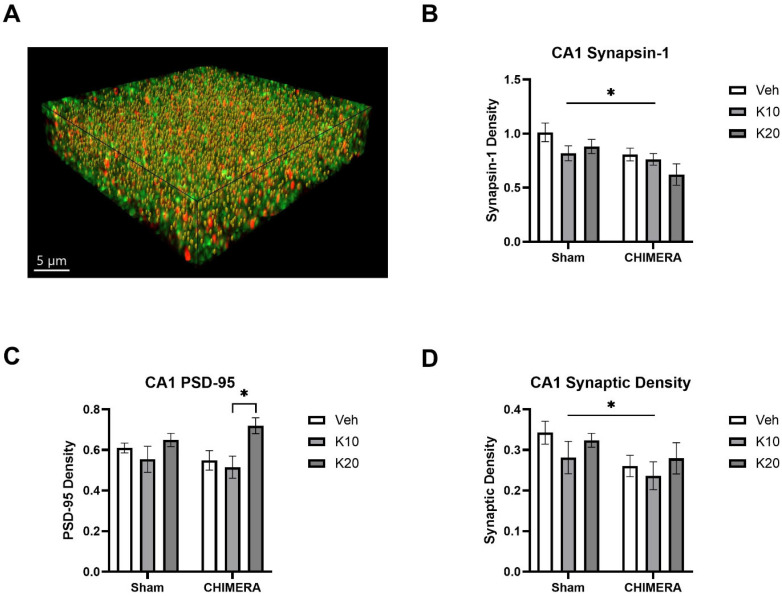
Synaptic density in the CA1 region of the hippocampus following CHIMERA injury and IV ketamine infusion in rats. (**A**) A representative Imaris image showing synapsin-1 (green), PSD-95 (red), and synaptic (gold) puncta. (**B**) Synapsin-1 density in the CA1. CHIMERA injury significantly reduced synapsin-1 density in the CA1 (* *p* < 0.05). (**C**) PSD-95 density in the CA1. Post hoc tests revealed a significant difference between K10 and K20 in CHIMERA animals (* *p* < 0.05). (**D**) Synaptic density in the CA1. CHIMERA injury reduced synaptic density in the CA1 region compared to the sham group (* *p* < 0.05). N = 9–10 per group.

## Data Availability

Data will be available upon reasonable request to the corresponding author.

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
