# Peer review of "Effects of Subanesthetic Intravenous Ketamine Infusion on Stress Hormones and Synaptic Density in Rats with Mild Closed-Head Injury"

_biomedicines, 2025, doi:10.3390/biomedicines13040787_

Round 1

Reviewer 1 Report

Comments and Suggestions for Authors

The goal was to test whether closed head mild TBI (CHIMERA) and ketamine affect plasma stress hormone levels and synaptic density in the mPFC and CA1 region of the hippocampus in rats. The authors report modest effects of ketamine on plasma corticosterone and progesterone and synaptic density in mPFC and CA1 of mice. They also report a modest effect of CHIMERA on postsynaptic density in the CA1.

Methods - The authors provide mg/kg dosage for ketamine. For i.v. administration with infusion pump, please also provide the volume, concentration, and flow rate. 

If tissue was harvested on PID4, why did the authors stop collecting behavioral data 2 hours post injury and stress hormone data 2 days post injury? What is the relevance of photobeam breaks during the two hours after ketamine infusion? The authors missed an opportunity to provide context to their PID4 histological data. This is a significant weakness.

Figure 2B and 2C and 3B and 3C - These data should be combined and presented as line graphs. Were data not collected prior to the first hour of infusion? Without pre-infusion values, the data are hard to interpret. These data should be analyzed using a mixed between-groups and repeated-measures (1st vs 2nd hour) ANOVA.

Results

Lines 251-252 - Any effect of CHIMERA on CORT is unknown given the lack of pre-injury baseline measurement.

Lines 261-264 - There are no effects of CHIMERA on CORT at any time in this study. The effects alluded to before are speculative without a pre-injury measure and the difference shown in Figure 2A was not statistically significant. 

Figure 3 - The graph should be formatted the same as Figure 2. 

Lines 370-372 - This speculation is not supported by data since there is no vehicle group.

Discussion

Lines 502-503 - What is the relevance of locomotor activity during ketamine infusion? In clinical situations a patient would not be locomotive.

Lines 507-508 - This is not true for PFC. 

Author Response

Methods - The authors provide mg/kg dosage for ketamine. For i.v. administration with infusion pump, please also provide the volume, concentration, and flow rate.

>> The infusion rate was calculated as 2.5 mL/kg/hr. Ketamine was diluted from 100 mg/mL stock solution with 0.9% sterile saline to 2 mg/mL for 10 mg/kg concentration  or 4 mg/mL for 20 mg/kg concentration (page 9). 

If tissue was harvested on PID4, why did the authors stop collecting behavioral data 2 hours post injury and stress hormone data 2 days post injury? What is the relevance of photobeam breaks during the two hours after ketamine infusion? The authors missed an opportunity to provide context to their PID4 histological data. This is a significant weakness.

>> We have studied behaviors until post-injury day 4 and published data in our previous paper (Spencer et al., 2023). We investigated motor activity, acoustic startle, and pre-pulse inhibition, and found impaired motor balance in the CHIMERA group in that study. In the current study, we used the same brain tissue samples collected at post-injury day 4 from the previous study, and we clarified that point in our revision (page 8).

>> The infrared photobeam breaks during the ketamine infusion reflect real-time locomotor activity exhibited by animals in the chambers. This serves several purposes, such as validation that rats are receiving correct doses of ketamine; 10 mg/kg infusion generally induces a sedative effect while 20 mg/kg infusion usually induces delayed hyperlocomotion in the second hour of the infusion owing to dissociative effects as we reported before (Radford et al., 2022; Spencer et al., 2023). Abnormal locomotor activity may also indicate poor health condition such as sickness and injury effects of animals.

Figure 2B and 2C and 3B and 3C - These data should be combined and presented as line graphs. Were data not collected prior to the first hour of infusion? Without pre-infusion values, the data are hard to interpret. These data should be analyzed using a mixed between-groups and repeated-measures (1st vs 2nd hour) ANOVA.

>> Thank you for the suggestion. We now combined Figure 2B and 2C to show time course effects of ketamine on CORT levels at 0 hr and 2 hr post-infusion. We reanalyzed the data using a mixed model with between subjects factors (CHIMERA and ketamine) and a within subjects factor (time) as suggested. There were significant main effects of ketamine F (2, 47) = 34.38, p < 0.0001 and time F (1, 46) = 287.9, p < 0.0001 as well as an interaction between ketamine and time F (2, 46) = 64.12, p < 0.0001. We also combined Figure 3B and 3C for PROG data the same way. We reanalyzed data using a mixed model with between subjects factors (injury and ketamine) and within subjects repeated measures (time). There was significant main effect of time F (1, 47) = 20.75, p < 0.0001. We used bar graphs instead of line graphs because GraphPad Prism software handled bar graphs better with this particular study design with 2 x 2x 3 groups. We updated Figure 2 (CORT) and Figure 3 (PROG) using these new graphs and also updated results section in page 14.

>> CORT levels at a baseline can vary significantly across animals due to many factors such as handling of animals, transportation between rooms, noise, lighting condition, etc. We previously found that pre-injury CORT levels fluctuate due to these factors and therefore we decided to only measure stress hormone levels after CHIMERA injury and ketamine infusion. After CHIMERA injury, animals were kept in a quiet and dark room for one hour undisturbed before drawing blood from jugular vein catheters. Also, during 2 hours of IV ketamine infusion in the chambers, animals were undisturbed which contributed to stable and clean levels of CORT and PROG. For comparison, we always included the control group: sham group for CHIMERA injury effects and saline vehicle group for ketamine effects.

Results

Lines 251-252 - Any effect of CHIMERA on CORT is unknown given the lack of pre-injury baseline measurement.

>> We agree with the reviewer. Although there was a trend of CORT increase in CHIMERA group compared to sham controls, it was not statistically significant (p=0.094). This may be due to relatively mild effects of CHIMERA in a single session used in the current study. We removed the graph from Figure 2 and just reported statistics in the results section (page 13).

Lines 261-264 - There are no effects of CHIMERA on CORT at any time in this study. The effects alluded to before are speculative without a pre-injury measure and the difference shown in Figure 2A was not statistically significant.

>> We agree with the reviewer and we changed our interpretation of data. We mentioned the reason for not measuring pre-injury CORT levels above. We removed CORT data at 1 hour post-injury from the Figure 2 and reported statistics in the results section.   

Figure 3 - The graph should be formatted the same as Figure 2.

>> Thank you for the comments. We now changed Figure 3 to be consistent with Figure 2.

Lines 370-372 - This speculation is not supported by data since there is no vehicle group.

>> As discussed in our methods section, rats received saline vehicle as a control compared to ketamine doses (10 or 20 mg/kg, IV).

Discussion

Lines 502-503 - What is the relevance of locomotor activity during ketamine infusion? In clinical situations a patient would not be locomotive.

>> Thank you for your comment. We used spontaneous locomotor activity monitoring to validate IV ketamine infusion effects in rats. Depending on doses, ketamine can produce either sedation (reduced activity) or dissociation (increased activity). These changes in behaviors can be easily detected by infrared photobeams installed in infusion chambers. We previously reported consistent findings on locomotor activity changes with IV ketamine infusion in rats. (Radford et al., 2018; Radford et al., 2022; Radford et al., 2020).Thus, spontaneous locomotor activity monitoring can serve as a positive control for ketamine infusion experiments in freely-moving rats.

Lines 507-508 - This is not true for PFC.

>> We have specified that we were referring to the CA1, which sustained damage after mTBI.  

Reviewer 2 Report

Comments and Suggestions for Authors

This manuscript by Boese et al, entitled "Effects of Subanesthetic Intravenous Ketamine Infusion on Stress Hormones and Synaptic Density in Rats with Mild Closed-Head Injury, reports a rigorous, methodically well-conceived study on neurophysiological influences of ketamine in rats who had suffered minor mTBI. The stress hormones and neuroplastic adaptations brought about by the infusion with subanesthetic doses of ketamine are seen in the scope of mTBI, considered one of the most common as well as grossly understudied conditions. The use of the CHIMERA model in conjunction with intravenous infusion of ketamine, coupled with advanced imaging techniques such as SEQUIN for synaptic quantification of the experimental design employed by the authors, reveals significant evidence of a dose-dependent synaptogenic effect of ketamine in the medial prefrontal cortex mPFC, postulated to be possibly with antidepressant and neuroprotective effects and also the attenuation of hippocampal synaptic density with mTBI. The biphasic corticosterone response to ketamine is interesting and worth further exploration into its long-term consequences. While the study is quite thorough, a more detailed time-course analysis and the inclusion of female subjects would increase the translational value of the work. Furthermore, behavioral tests would make the link between synaptic changes and functional consequences even stronger. I must say, the manuscript is well written and contributes valuable insights into the field of neurotrauma and psychopharmacology. Acceptance with some minor revisions if suggested by other reviewers is recommended. Well done!

Author Response

This manuscript by Boese et al, entitled "Effects of Subanesthetic Intravenous Ketamine Infusion on Stress Hormones and Synaptic Density in Rats with Mild Closed-Head Injury, reports a rigorous, methodically well-conceived study on neurophysiological influences of ketamine in rats who had suffered minor mTBI. The stress hormones and neuroplastic adaptations brought about by the infusion with subanesthetic doses of ketamine are seen in the scope of mTBI, considered one of the most common as well as grossly understudied conditions. The use of the CHIMERA model in conjunction with intravenous infusion of ketamine, coupled with advanced imaging techniques such as SEQUIN for synaptic quantification of the experimental design employed by the authors, reveals significant evidence of a dose-dependent synaptogenic effect of ketamine in the medial prefrontal cortex mPFC, postulated to be possibly with antidepressant and neuroprotective effects and also the attenuation of hippocampal synaptic density with mTBI. The biphasic corticosterone response to ketamine is interesting and worth further exploration into its long-term consequences. While the study is quite thorough, a more detailed time-course analysis and the inclusion of female subjects would increase the translational value of the work. Furthermore, behavioral tests would make the link between synaptic changes and functional consequences even stronger. I must say, the manuscript is well written and contributes valuable insights into the field of neurotrauma and psychopharmacology. Acceptance with some minor revisions if suggested by other reviewers is recommended. Well done!

>> Thank you for your comment. Our previous study (Spencer et al., 2023) found limited behavioral deficits following this CHIMERA injury paradigm. The one example was a rotarod performance on day 3, which may be relevant to hippocampal function due to vestibulomotor function. Future studies should examine functional significance of mild TBI and ketamine on behavioral indices of major depression, anxiety, and fear/stress.

>> We also explained the rationale behind the four day time point we selected. The four day post-injury timepoint was chosen because it is a significant time point for synapse functionality and maturation, where synapses surviving past four days become persistent (Knott et al., 2006) as well as for the time course of synaptic proteins such as PSD-95 and synapsin-1 after TBI (Ansari et al., 2008). Stress hormone data were analyzed in the acute period only (one day post-injury and infusion) because we expected that the changes would be transient, as CORT peaks and troughs in the few hours after stress exposure (Spencer & Deak, 2017), as well as the short half-life of ketamine (several hours). However, the fact that progesterone levels were significantly elevated at one day after CHIMERA, further studies are warranted for longer time points post-injury.

>> Also, future studies should include female rodents in this context. Other studies reported sex differences in synaptic effects of ketamine between male and female rodents (Sarkar & Kabbaj, 2016; Thelen et al., 2019) and we would expect that female rodents may show different synaptic responses to ketamine and mild TBI.

Round 2

Reviewer 1 Report

Comments and Suggestions for Authors

The authors addressed almost all of my concerns.